# Green Roof Design: State of the Art on Technology and Materials

**Stefano Cascone** 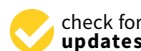

Department of Civil Engineering and Architecture, University of Catania, Via Santa Sofia 64, 95123 Catania, Italy; stefano.cascone@unict.it

**Abstract:** In order to consider green roofs as an environmentally friendly technology, the selection of efficient and sustainable components is extremely important. Previous review papers have mainly focused on the performance and advantages of green roofs. The objective of this paper is to examine the primary layers: The waterproof and anti-root membranes; the protection, filter, and drainage layers; the substrate; and the vegetation. First, the history, modern applications, benefits and classification are analyzed in order to present a well-defined state of the art of this technology. Then, the roles, requirements, characteristics, and materials are assessed for each green roof layers. This technology was compared to a conventional roof technology, Mediterranean climate conditions and their influence on green roof design were assessed, also comparing them with Tropical area and focusing on irrigation systems, examples about the commercial materials and products available in the market were provided and innovative materials coming from recycled sources were analyzed. Future research should evaluate new materials for green roof technologies, in order to enhance their performance and increase their sustainability. The information provided in this review paper will be useful to develop Mediterranean green roof guidelines for selecting suitable components and materials during the design and installation phases.

**Keywords:** conventional roof; Mediterranean climate; irrigation systems; commercial products; international market; recycled components

---

## 1. Introduction

Currently, climate change and the scarcity of natural energy resources are topics of interest in many countries [1]. Furthermore, cities continue to grow and expand their peripheries to accommodate increases in rural migration to urban areas. According to a recent report of the United Nations, urbanization is forecasted to attain 83% by 2030 in developed countries [2]. This results in several environmental issues on a global scale, such as increased greenhouse gas emissions.

Due to this worldwide urbanization, the demand for new buildings, land, water, and energy have drastically increased over the last four decades. According to United Nations Environmental Program, the construction and maintenance of buildings account for about 40% of the global primary energy requirement and buildings account for 33% of the global greenhouse gas emissions [3]. Therefore, the building sector is of particular interest in the reduction of energy use, in order to limit global warming and mitigate the impacts of climate change [4,5].

The effect of building envelope technologies on the design and construction of sustainable buildings and urban spaces is undeniable [6,7]. Implementing various sustainable approaches and designing more environmentally friendly components for buildings leads to the realization of low-energy buildings [8]. In addition, roofs are important components of buildings, accounting for nearly 20–25% of the overall urban surface area [9]. Therefore, efficiently designed and integrated green roofs have great potential to affect the building and urban environments, replacing the lost green spaces and

habitats in modern cities. Specifically, green roofs are engineered roofing systems, planted with different kind of plants on the top of a growth medium [10].

In recent years, the number of studies carried out on green roofs has considerably increased and several review papers have been published, in an attempt to summarize and organize the scientific knowledge on this topic. One of the first reviews was carried out in 2010 by Berndtsson [11], which addressed the role of green roofs in urban drainage, considering the management of both water quantity and quality. Factors which affect the influence of a vegetative roof on runoff water quality were discussed in general terms, followed by a review of data regarding the concentrations of phosphorus, nitrogen, and heavy metals in the runoff, pH, and the first-flush effect. Likewise, Akther et al. [12] statistically synthesized the effects of the influential factors, including design and hydrologic variables, on green roof performance, to explore their effects in different climatic zones. Castleton et al. [13] reviewed the current literature and highlighted the situations in which the greatest building energy savings can be made. Similarly, Saadatian et al. [14] focused on energy-related topics. Berardi et al. [15] presented a state-of-the-art of green roofs emphasizing current implementations, technologies, and benefits. The authors reviewed the benefits related to the reduction of building energy consumption, mitigation of the urban heat island effect, improvement of air pollution, water management, an increase of sound insulation, and ecological preservation. In 2015, Hashemi et al. [16] provided an overview of the effects of the application of the green roof strategy on the quality of runoff water and the reduction of energy consumption. Shafique et al. [17] included in their review the history, components, and multiple benefits (environmental, social, and economic) associated with green roof technology. In addition, the authors also emphasized its performance in reducing stormwater and energy costs, improving air quality, and ecological benefits. Recently, Cascone et al. [18] carried out a comprehensive review of the cooling effect, due to the evapotranspiration process, as most of the benefits of green roofs are related to this phenomenon. These previous studies were mainly focused on reviewing the performance and benefits, without providing a description of their technology and materials.

Vijayaraghavan [19] analyzed desirable characteristics for the growth substrate and vegetation and suggested a methodology for constructing green roofs. Dvorak and Volder [20] conducted a review in order to investigate what is known about the application of plants on vegetative roofs across North America and their ecological implications. However, these review papers addressed mainly the roles and the performance of vegetation and substrate, providing little information with both of the materials and of the other primary layers, such as the waterproof and anti-root membranes, and the protection, filter, and drainage layers. In addition, the most used international guidelines are the German FLL 2018 [21], concerning the planning, construction and maintenance of green roofs. However, these guidelines are mainly developed for Northern Europe, characterized by cold and rainy days during most of the year. Mediterranean area, characterized by hot and sunny days, has requirements that are not fulfilled by the green roof designed, according to the German FLL standard. This is mainly due to the absence of water for extended periods. Actually, several regional guidelines exist. For example, in Italy, the guideline is the standard UNI 11235:2015 [22]. However, this standard is written in Italian and, therefore, it is not suitable as an international guideline for the Southern Europe countries.

Differently from both the review carried out by Vijayaraghavan [19] and the international FLL guidelines [21], the novelty of this paper consists in comparing it to a conventional roof technology, in terms of both materials and thermal and economic performance, in assessing the Mediterranean climate conditions and their influence on green roof design, also comparing it with Tropical area and focusing on irrigation systems, in providing examples about the commercial materials and products available in the market and in analyzing innovative materials coming from recycled sources, as possible components. All these aspects related to green roof materials and technology are not fully described neither by previous articles nor by international guidelines. In addition, for each layer, the roles, requirements, performance, and materials are assessed. The information provided in this review paper will be useful for both researchers and designers to develop Mediterranean guidelines for selecting suitable components and materials during the design and installation phases.

First, the history and modern applications are discussed, in order to present a state of the art of this technology and their benefits and classification into extensive and intensive are described.

## 2. History and Modern Applications

Existing literature shows that covering the building rooftop with soil, wetting the soil, and shading the surface of the wet soil have been used for centuries as passive cooling practices in different countries with confirmed benefits in different climatic conditions and building characteristics [23].

One of the most famous ancient green roofs dates to the fifth century when the Hanging Gardens of Babylon was constructed that is admitted as the earliest examples of greenery systems [24]. Living roofs were also utilized in the ziggurats of ancient Mesopotamia. Like Babylon, the Roman and Greek architecture also employed these systems at their own eras. For example, the Mysteries Villa represents such integration and offers an example of space that enhances human activities while improving the aesthetic value and roof life. In the Mediterranean region, different plants notably vines were utilized to prevent the building envelope from excessive sunlight in the summertime and to provide cooler and comfortable indoor conditions to occupants. Green roofs have also been presented in vernacular architecture in different countries. For example, the usage of the plants climbing the building greatly expanded in the UK and Central and Northern Europe (especially in Norway) during 17th and 18th centuries to increase the thermal insulation [25]. After many centuries of rare utilization in European cities, during the modern age green roofs have been rediscovered in the twentieth century by the Swiss architect Le Corbusier who included them in the five points of modern architecture [26]. Around the same time, American organic architects proposed vegetative roofs as a method to integrate buildings and nature.

Modern green roofs, therefore, may acquire their concept from ancient technique; however technological advances have made this technology far more efficient, practical and beneficial than their ancient counterparts. An intensive implementation started from Germany in the early 1960s when there were energy crises arose [27]. Several investigations have been carried out with emphasis on biodiversity, substrate, roof construction and design guidelines. Green roofs gained popularity also in Austria, Switzerland and United Kingdom (UK) in the same years, however, Germany is regarded as the world leader in the employment of this strategy, because green roofs on the large scale were being developed, designed and implemented [28]. In this respect, the first comprehensive program was put into practice from the early 20th century by retrofitting the houses with greenery surfaces. Nowadays, as Table 1 shows, research and application at the building in Germany are very popular and green roof coverage increases by approximately eight million square meters per year, which is remarkable. The total value of this technology in Germany was estimated to be worth Eur 254 million in the year 2015.

**Table 1.** Estimated market figures [29].

| Target Country | Green Roof Total m$^2$ (2014) | Green Roofs New/Year m$^2$ | Ratio Extensive | Ratio Intensive | Yearly Sales Figures € |
|---|---|---|---|---|---|
| Austria | 4,500,000 | 500,000 | 73% | 27% | 27,350,000 |
| Germany | 86,000,000 | 8,000,000 | 85% | 15% | 254,000,000 |
| Hungary | 1,250,000 | 100,000 | 35% | 65% | 5,662,500 |
| Scandinavia | - | 600,000 | 85% | 15% | 16,050,000 |
| Switzerland | - | 1,800,000 | 95% | 5% | 51,300,000 |
| United Kingdom | 3,700,000 | 250,000 | 80% | 20% | 28,000,000 |

## 3. Green Roof Benefits

Green roofs are a solution to increase the sustainability and energy conservation of buildings, but they produce several other benefits to urban areas in terms of social, economic, and environmental advantages. Some of these benefits can be illustrated as reducing greenhouse gas emissions and

the urban heat island effect [30], preventing acid rain by escalating pH values [31], improving air quality [32] by producing more oxygen and sequestering carbon dioxide and decreasing traffic noise pollution within urban areas [33]. Other benefits of green roofs are the enhancement of aesthetic value in urban environments and the improvement of life quality of dwellers by creating recreational activities [34].

Several studies stressed the advantages for urban hydrology and storm water management, focusing on the ability of green roofs to minimize the risks of flooding by reducing water runoff while improving its quality [35]. As a result of this improvement, due to the absorption of rainfall in the soil, the burden on water treatment facilities is reduced.

Green roofs can reduce the sound exposure near or inside a building by mitigating diffracting sound waves over (parts of) roofs and by reducing sound transmission through the roof system [36]. Commonly used porous growing substrates were shown to have good sound absorbing properties when dry [37].

Based on the current literature, the energy-related performance of green roofs is still the most common benefit for which they are promoted and adopted [38]. Therefore, energy designers are very interested in their application, due to the reduction in roof surface temperature and solar heat to the covered building components [39], highlighting their contributions to both overall building thermal performance and microclimatic conditions in urban environments [40,41]. Green roofs improve the thermal performance of a building through different mechanisms:

- Shading: Vegetation provides an additional layer that shades the substrate and the roof, blocking part of the incoming solar radiation;
- Evapotranspiration: Plant transpiration and soil evaporation cool the surface of the plants, decreasing the heat flux toward the interior of the building and the urban heat island effect;
- Thermal inertia: The substrate increases the roof thermal mass, delaying and reducing incoming heat fluxes;
- Thermal insulation: The substrate and drainage layers increase the heat resistance of the roof by providing an additional thermal layer.

Therefore, this strategy is a sustainable roof design that saves energy for cooling and heating purposes [42]. Despite its high initial cost, in the long term, green roofs are an economical option considering their energy savings. However, the focus of developers has been limited to achieving basic aesthetic benefits. This is generally due to a lack of research on different aspects of vegetative roofs and the premature introduction of products into the market [43].

## 4. Technology Classification

Green roofs are broadly classified into intensive and extensive roofs, though some authors include a semi-intensive classification, based on the depth of the substrate layer, maintenance, cost, vegetation type, construction material, and irrigation [44].

Intensive green roofs are generally roof gardens designed with a considerable substrate depth—more than 15–20 cm—a wide variety of plants (similar to ground-level landscapes), high water retention capacity (over 50%), high capital costs ($25 per square foot), and heavy weight (180–500 kg/m$^2$). Typically, this type is installed when the slope is less than 10°. Due to the increased soil depth, the plant selection can be more diverse, including small trees, shrubs, and bushes [24,45]. Therefore, it requires a high level of maintenance, in the form of fertilizing, weeding, and watering. One of the main advantages of an intensive roofing system is the creation of a natural environment with improved biodiversity, providing a recreation space, as they are normally designed for the use of humans for entertainment [46]. Intensive roofs encompass a comparatively better potential than extensive green roofs in terms of stormwater management, decreasing runoff by 85% when compared to traditional roofs [47]. Likewise, intensive green roof runoff has three times less lead contamination, 1.5 times less zinc contamination, 2.5 times less cadmium contamination, and three times less copper contamination [47]. On the other

hand, their greater weight may require additional structural reinforcement, and drainage and irrigation must generally be utilized, increasing the technical complexity and associated costs [27].

Extensive green roofs are characterized by a shallower depth of substrate layer (less than 15 cm) and have a lower weight in comparison to intensive ones. Owing to the thin substrate layer, extensive roofs can utilize only limited types of plants, including grasses, mosses, and a few succulents. The main advantages of extensive roofing systems are the low capital cost and maintenance and water requirements, compared to intensive roofs [11]. These roofs are usually very lightweight and useful, especially where no additional structural support is desired. Furthermore, an extensive roof can be installed on a larger slope, their construction process is technically simple, and it is appropriate for large-sized rooftops. However, both the energy performance and storm water management potentials of extensive green roofs are relatively low [48].

Of the two types, extensive roofs are most common around the world, due to their low weight, not requiring irrigation, and having less capital and maintenance costs [49]. Table 2 compares intensive and extensive green roofs.

**Table 2.** Main features of intensive and extensive green roofs.

| Main Characteristics | Intensive | Extensive |
|---|---|---|
| Maintenance | High | Low |
| Irrigation | Periodically | Regularly |
| Plant diversity | Sedum-Herb-Moss-Grass | Lawn-Perennial-Shrub-Tree |
| Cost | Low | High |
| Weight | Lightweight (60–150 kg/m$^2$) | Heavy (180–500 kg/m$^2$) |
| Thickness | 60–200 mm | 140–400 mm |
| Use | Accessible | Inaccessible |

## 5. Materials and Components

A green roof generally consists of several components, including, from bottom to top [19]: A waterproofing membrane, an anti-root barrier, a protection layer, a water storage and drainage layer, a filter layer, substrate (growing medium or soil), and vegetation (plants). These components are shown in Figure 1.

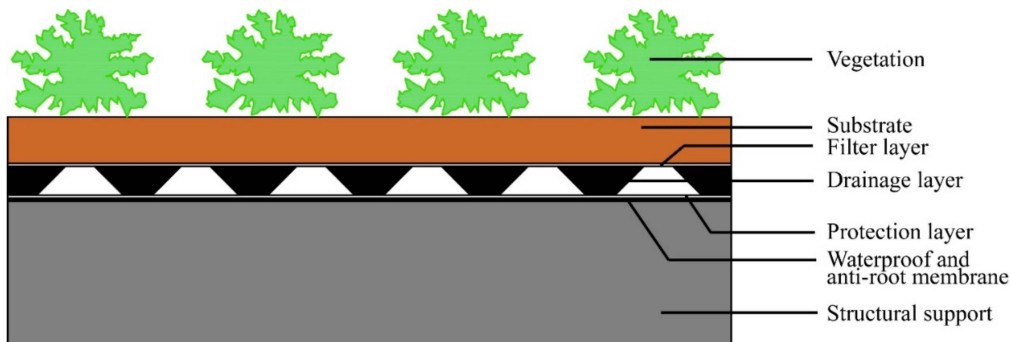

**Figure 1.** Green roof layers.

### 5.1. Waterproof Membrane

The waterproof membrane is one of the most important components of the green roof technology. It protects the building against any infiltration resulting from the large water content of the upper layers and, in turn, it is protected by the vegetative roof against temperature fluctuations and solar radiation, which may cause performance decay of the membrane in a short time. The primary requirement for this layer is water-tightness. In addition, it should be considered that the maintenance of this layer is very complex because, in the case of a leak in an operating green roof, all the layers need to be

removed. Therefore, it is necessary to foresee solutions to prevent horizontal water flow below the membrane, to reduce degradation, and to allow the location of any infiltration points. These results can be achieved either through the perfect adhesion of the waterproof membrane to the bearing structure, or through compartmented sectors in the membrane.

The design of the waterproofing membrane is similar to traditional roofing. However, compared to a traditional roof, the waterproofing membrane of a green roof is protected against UV rays, thermal fluctuations, and hail shocks. On the other hand, the membrane can be exposed to the biological and chemical agents contained in the substrate and vegetation.

Bituminous flexible membranes are the most common and they can be classified, further, as:

- Elastomeric membranes: Characterized by an elastomeric polymer mixed with bitumen, which gives flexibility at low temperatures and excellent elasticity;
- Plastomeric membranes: Characterized by a plastomeric polymer mixed with bitumen, which gives stability at high temperatures and offers high resistance to UV exposure;
- Elasto-Plastomeric membranes: Combines the characteristics of the two membranes above-described.

The bituminous membranes can be laid in a monolayer or as a double layer. Thicknesses of 3 or 4 mm are generally used. These membranes have different characteristics and behaviors, but a common stratigraphy is realized by the compound, the glass or polyester reinforcement, and the protective surface finish.

The main characteristics of the waterproofing membrane to be controlled, in addition to water tightness, are the dimensional stability (since as long as the green roof is not installed the membrane is exposed to solar radiation and to high daily thermal fluctuations), cold flexibility, resistance to static loads (in order to verify that the membrane resists permanent and accidental loads), and artificial ageing (through long-term exposure to high temperatures). The waterproofing membrane does not necessarily meet the requirement of protection from the roots. If the membrane is exposed to the roots and there is no anti-root layer, it would be necessary to verify this resistance, too.

The type of green roof, along with cost, availability, and life expectation, determines the type of waterproofing. Once the suitable typology of waterproofing membrane has been identified, it must be laid on a sloping screen (allowing an adequate flow of water) and turned over at the edges by at least 15 cm.

## 5.2. Anti-Root Membrane

In the design of a green roof, the aggressive capacity of the root system should not be underestimated. The objective of a root-barrier is to defend the waterproof membrane and the roof structure from vegetative roots penetrating from the upper layers, which could mechanically disrupt and chemically alter the waterproofing membrane. The consequence of these two combined actions is the drilling of the waterproofing membrane and penetration into the underlying layers, causing water infiltrations in the building. Therefore, the anti-root layer must always be laid out in a green roof and, in almost all cases, it is integrated into the waterproofing membrane. When installing a living roof on an existing building with an efficient waterproofing membrane, the additional anti-root protection layer will be overlaid onto the waterproofing membrane.

The main characteristics and materials of this layer are similar to those reported for the waterproofing membrane. However, the anti-root membrane must be characterized by high resistance to micro-organisms contained in the soil, adding some repellent ingredients to the chemical composition of the anti-root membrane. Anti-root membranes are characterized by thicknesses around 4 mm and are placed with hot-air welding or a chemical solvent.

The use of concrete as an anti-root barrier is not possible, as it can be attacked by roots over time and makes it very difficult to maintain the waterproofing membrane. In addition, felts, polyethylene films, or the like, do not meet the performance requirements in terms of resistance to root penetration. An alternative material for the anti-root layer is a metal sheet.

## 5.3. Protection Layer

To protect the waterproofing and anti-root membrane, a good practice is to provide a separation and protection layer among the green roof layers. The requirement for this layer is the ability to withstand loads and stresses during both the construction and operational phases. Therefore, it is necessary to place it after the anti-root membrane. Normally, the loads it needs to withstand are, due to the weight of the layers above the anti-root membrane.

The materials used are either geogrids and geotextiles or polystyrene, with a minimum thickness of 3 mm and compression resistance >150 kPa. These materials may not replace the anti-root membrane.

Some materials used for the protection layer could accumulate water, which can be released to the vegetation during periods of drought.

## 5.4. Water Storage and Drainage Layer

The drainage layer plays a crucial role in the correct development of a green roof. As most of the vegetation needs a ventilated and non-waterlogged substrate, this layer aims to drain the excess of water from the substrate, allowing a suitable equilibrium between air and water and providing adequate ventilation for the roots. In addition, by evacuating extra water, it decreases the load on the building structure and reduces the risks of a mechanical breakdown. Furthermore, the drainage layer defends the waterproof membrane and enhances the thermal performance.

Among all these roles, the aeration of the root apparatus is often penalized by incorrect green roof design, resulting in the failure of the entire technology. Therefore, under operating conditions, the drainage layer should be filled, at least 60%, with air, to preserve the vegetation and prevent deterioration. The draining and ventilation capacity of the drainage layer could decrease over time, influencing the development of the vegetation [50].

There are two main materials used for the drainage layer (Figure 2):

- Granular materials: These have a minimum thickness of 6 cm and a minimum density of 150 kg/m$^3$. If porous, they are also used as water storage. The main aggregates used in green roofs are pozzolana, pumice, lapilli, expanded clay, expanded pearlite, expanded slate, and crushed bricks;
- Modular panels: These have a thickness between 2.5–12 cm and a weight of about 20 kg/m$^2$. These panels are produced with high-strength synthetic or plastic materials (polyethylene or polystyrene) and cavities to store water while still allowing the removal of surplus water.

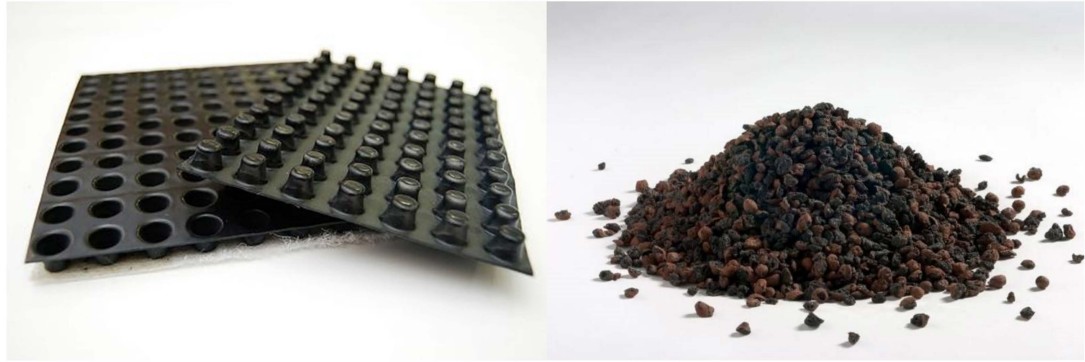

**Figure 2.** Green roof drainage materials: Modular panels (on the left) and granular materials (on the right).

For granular materials, the requirements are permeability (i > 0.02), compressive strength (>1.5 N/mm$^2$), and thermal conductivity ($\lambda$ < 0.2 W/mK). For modular panels, the required characteristics are vertical draining capacity under operating loads (0.7 l/m$^2$s), compressive strength, having longitudinal tensile strength (>7.0 kN/m) and tensile strength (>7.0 kN/m to be applied only for

green roofs with slope >30%), and resistance to aggressive agents. These panels are made with cavities to store water during rain and make it available during drought periods. From these cavities, the water evaporates and penetrates the filter layer where it condenses, reaching the root apparatus by capillarity.

The selection of a suitable drainage layer varies greatly according to the rainfall characteristics of the site, construction needs, structural requirements, costs, green roof size, roof slope, quantity and flow of discharges, and plant species. Moreover, the choice of the drainage layer depends on the hydraulic flow and the vertical load, since, during the operative phase, there is either a compaction (for granular materials) or a deformation (for plastic materials).

Generally, for small-scale green roofs, as in residential buildings, granular materials meet the requirements. Nevertheless, an important disadvantage of granular materials is that they can only be used on flat or slightly sloped roofs (<5°). In addition, limitations during installation and workmanship cannot be ignored. Modular panels are used for larger green roofs and, compared with granular material, they have higher performance thanks to their reduced thicknesses and weights, better management of the air/water ratio, ability to store more water in their cavities, greater mechanical resistance, and increased durability. In addition, transport, handling, installation, and maintenance (with the possibility of opening up the plastic modules as pieces) are simpler and quicker. On the other hand, cost and disposal are the major limitations of plastic drainage modules.

### 5.5. Filter Layer

The key role of this layer is to separate the substrate from the drainage layer and to avoid smaller particles (for example fine soils and vegetation debris) entering and clogging the drainage layer, thus reducing its performance over time. This material, once penetrated, could favor the establishment of plants inside the drainage layer or obstruct the drains, causing infiltrations and blocking the entire greening system. The filter layer should have small holes to allow for high water permeability, at least 10 times higher than the substrate. Therefore, the performance to be monitored is the water permeability.

The two types currently used for the filter layer are:

- Granular materials, such as pozzolana, pumice, lapillus, expanded clay, expanded perlite, expanded slate, and crushed bricks, characterized by a water permeability greater than 0.3 m/s;
- Non-woven geotextiles with water permeability greater than 0.3 cm/sl $\times 10^{-3}$ m/s, able to absorb 1.5 L/m$^2$ of water.

Generally, geotextile materials are used for the filter layer.

The main parameters required for the filter layer are to withstand the weight overhead and punching resistance (>1.100 kN), longitudinal tensile strength (>7.0 kN), transverse tensile strength (>7.0 kN), deformation to the longitudinal operating load (<60%), deformation to the transverse working load (<60%), effective pore opening (0.10–0.20 mm), oscillation resistance (<20%), and resistance to aggressive agents.

### 5.6. Substrate

The growing media should be designed to accomplish the numerous long-term advantages of a green roof, such as water quality improvement, peak flow decrease, and noise and thermal insulation, which are related to the substrate characteristics. The substrate plays an important role in the plant growth, as it guarantees establishment and stability, satisfying the control of the agronomic capacity; that is, the capacity to maintain the physical, chemical, and biological conditions for the correct vegetative development.

The thickness and weight of the substrate depend on the vegetation, roof geometry, climatic conditions, and irrigation strategy. In the rain, some substrates become saturated rapidly, increasing their weight. Generally, substrate weight varies from 12–14 kg/m$^2$ with a thickness of 8 cm for extensive green roofs to about 600 kg/m$^2$ with a thickness of 50–60 cm for intensive ones.

The substrate is characterized by two main sets of parameters:

- Physical parameters, such as density, particle size, water permeability, maximum water volume, and maximum air volume in saturated conditions;
- Chemical parameters, such as pH index, electrical conductivity, and quantity of organic matter.

In the case of a wrong choice of substrate, the consequences are compaction, imbalances between water and air, asphyxia of the root apparatus, increased weight, reduction in drainage, and the alteration of the nutrients.

5.6.1. Performance

The main characteristics required for the substrate in the development and maintenance of vegetation under different climate conditions are:

- High hydraulic conductivity and water retention capacity;
- High aeration and flow attributes;
- Poor leaching and high sorption capacity;
- Lightweight, locally available, and cost effective;
- Stability of the physical and chemical structure in severe climate conditions;
- Minimum organic content;
- Wide variety of vegetation;
- Improved water quality.

Optimization of the substrate performance is achieved through a mixture of materials with different characteristics and proportions. In this framework, it is very difficult to find or formulate a green roof substrate with all these beneficial attributes, since some qualities could be decreased to enhance others, depending on the performance required. For example, a low-bulk density attained by employing lightweight materials could compromise the stability of the substrate and vegetation anchorage. Furthermore, by reducing the particle size and increasing the organic matter content to improve water retention capacity could influence air-filled porosity and hydraulic conductivity. Therefore, enhancing these attributes through scientific investigation is necessary for long-term sustainability.

Green roof substrates should be characterized by low dry and wet bulk densities, as they represent the main load on the roof bearing structure, especially in old buildings where the roofs were not built to accommodate green roof systems [51]. One of the key approaches for decreasing the weight of the substrate is to utilize low-density inorganic materials. The bulk density of perlite was stated to be 9.4 times less than that of conventional garden soil. It should also be noted that the lower the density of the substrate, the thicker the substrate can be constructed, and the larger variety of vegetation that can be planted.

Concerning the hydraulic performance, water retention and permeability should be considered, guaranteeing a porosity not less than 58% and 48% for extensive and intensive green roofs, respectively. Schultz et al. [52] found that 125 mm and 75 mm vegetative roofs retained 32.9% and 23.2% of all precipitation by volume, respectively. Farrell et al. [53] tested whether two different water-retention additives (silicate granules and hydrogel) increased substrate water retention capacity and plant available water. Two substrates were compared, one based on scoria and the other based on crushed terracotta roof-tiles. Without additives both substrates had similar water holding capacity (40–43%). Furthermore, Vijayaraghavan, and Joshi [54] prepared a substrate using 30% perlite, 20% vermiculite, 10% sand, 20% crushed brick, 10% cocopeat and 10% *T. conoides* was found to have high water retention capacity (49.5%). However, Talebi et al. [55] revealed that the vegetation type had a greater impact on the water retention performance of green roofs than increases in substrate storage capacity associated with different substrate depth, porosity and wilting point over the range assumed in this study.

In addition, the substrate increases the thermal resistance of the green roof. However, the substrate is not considered to be an insulating material, due to the variability in the water content, which

significantly influences the thermal conductivity. For this reason, the thermal performance of the green roof should be referred to at the maximum water saturation.

Numerous researchers examined the leaching tendency and sorption ability of green roof substrates, which affects the quality of the runoff. However, due to the percentage of inorganic components, the sorption ability of the substrate is reduced. For example, expanded perlite, a commonly-used substrate constituent, showed no more than 8.6 and 13.4 mg/g sorption abilities on Cu(II) and Pb(II) ions, respectively. An alternative broadly utilized substrate element, pumice, adsorbed only 3.5 and 1.6 mg/g of Cu(II) and Cr(III), respectively.

The water holding capacity (WHC) of the substrate components is essential for the endurance of the vegetation, since it delays the peak flow during storm events and helps the plants to withstand drought conditions. In addition, high WHC allows the use of non-succulent plant species. Forschungsgesellschaft Landschaftsentwicklung Landschaftsbau (FLL) [21] suggests WHC >20% for extensive green roofs. WHC can be improved by increasing the substrate volume, depth, and organic content. In recent years, some researchers have suggested the use of additives to maximize the water holding capacity of the growing media. Vijayaraghavan and Joshi [54] incorporated a brown seaweed (Turbinaria conoides) in the growth substrate to enhance the runoff quality of green roofs.

The ventilation and flow characteristics of the substrate are not only important for vegetative development, but in avoiding roof leakage and water overloading. For extensive roofs, FLL recommended an air-filled porosity >10% and hydraulic conductivity >3600 mm/h. Large-sized particles increase air-filled porosity and hydraulic conductivity, while small-sized particles and organic matters reduce air and flow features.

5.6.2. Composition

The common procedure is to blend various materials with different attributes at well-defined percentages to constitute the growth substrate. However, the substrate in green roofs differs from traditional garden soil, as traditional moulds are mainly composed of organic materials, such as peat and compost. The substrates consist mainly of mineral materials (Figure 3), varying from 50% to 90% of the substrate volume and giving the green roof a lower density, higher porosity (75%), higher draining capacity in saturated conditions, and easier ventilation of the roots. The most used low-density inorganic materials for the substrate are pumice, zeolite, scoria, vermiculite, perlite, peat, and crushed brick. In particular, the particle size should have a high percentage of granules with a diameter between 2–4 mm. Some researchers recommended the use of more than 80% inorganic materials in the composition of the growing medium and, thus, the load of green roofs can be lowered.

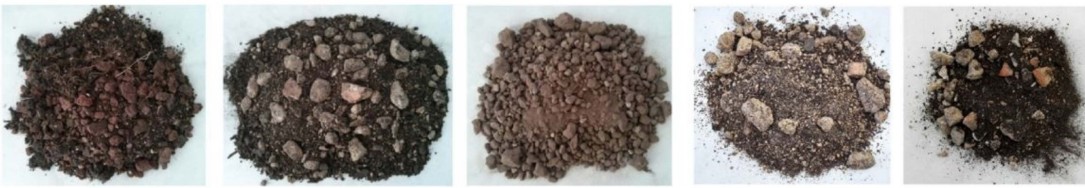

**Figure 3.** Commercial substrates analyzed by Coma et al. [56]

However, the growth medium is normally projected to incorporate nutrients to encourage vegetative development. Therefore, it is crucial to include organic components in the substrate to supply nutrients to the green roof. Some popular organic components used in the substrate are mulch, peat, and various other composts. Eksi et al. [57] quantified the optimal percentage of compost in a substrate for optimizing the growth and yield of cucumbers (*Cucumis sativus*) and peppers (*Capsicum annuum*). Cao et al. [58] examined the effects of adding one type of green waste, biochar, to two scoria-based substrates (with or without added organic matter) on bulk density. The results showed that biochar significantly reduced the bulk density, and substrates with 40% biochar had an

additional 1.5 cm/m$^2$ of depth (compared to the same weight of scoria), further increasing the available water for the plants and rainfall retention.

When commercial substrates are prepared for the expected plant species, climate conditions, and maintenance levels, they may not work properly in different geographic regions. Ziogou et al. [59] used commercially-available materials for green roof layers (Bauder enterprise), focusing both on energy conservation and sustainability in two alternative solutions applied to a typical urban office building in representative climatic areas of Cyprus, in the Eastern Mediterranean. Vijayaraghavan et al. [60] used a commercial substrate (Daku enterprise) to assess the runoff quality from green roofs.

*5.7. Vegetation*

Most of the success of a green roof depends on how healthy the plants are, especially in meeting long-term client expectations. The benefits mainly depend on the plant species, as they enhance both water and air quality and thermal performance. In addition, the vegetation characterizes the visual aspect of the green roof, prevents the erosion of the substrate, and provides protection for various animal species, especially arthropods and birds.

In the choice of vegetation, the climate conditions, such as rainfall intensity, humidity, wind, and solar radiation, should be considered. Furthermore, the substrate mixture also affects the plant species that can be installed, especially in terms of pH, salinity, and nutrients.

In recent years, some authors have worked to identify suitable plant species based on the soil depth, defining the following plant species for extensive green roofs:

- 0–5 cm: Sedum, mosses, and lichens;
- 5–10 cm: Short wildflower meadows, long-growing, drought-tolerance, perennials, grasses, alpines, and small bulbs;
- 10–20 cm: A mixture of low or medium perennials, grasses, bulbs and annuals from dry habitats, wildflowers, and hardy sub-shrubs.

The roof installation site is important in optimizing the choice of plant species, as the emissions of hot/cold air and the chemical components in the air should be considered. Moreover, the shaded areas (due to surrounding buildings) alter the solar radiation and, consequently, the luminous flux and temperatures on the green roof. However, the roof of the building is not a natural ecosystem for the development of plants, as water is a restricting element and its availability varies between rain events. In addition, building load constraints limit the substrate weight and depth and, therefore, restrict the types of vegetation that can be used.

5.7.1. Performance

The favorable attributes of plants for extensive green roofs are good ground coverage, short and soft roots, phytoremediation, ability to survive in extreme climate conditions and under minimal nutrients conditions, limited maintenance, and fast development. Even if it is very difficult to find plant species with all these valuable qualities, a considerable improvement has been achieved for the choice of proper vegetation. Ground coverage is a key criterion for plant choice as it shields the substrate from direct sunlight and wind. Furthermore, ground cover plants delay unwanted plant growth and soil erosion when constructed on sloped rooftops. Plants with short, soft roots avoid the infiltration of the roots into the roof deck. The growth medium also requires only minimal nutrients to avoid wildflowers and the production of eutrophic runoff and, thus, only needs nutrient-poor inorganic recycled constituents as the main components of the green roof substrate.

A capacity for phytoremediation has never been a standard for choosing the green roof vegetation. Plants eliminate dissolved contaminants by phytoextraction and vaporous pollutants by phytovolatilization. While there have been numerous experiments on air and water quality improvements by green roofs, the role of vegetation on contamination management has rarely been examined. Baraldi et al. [61] evaluated the potential ability of fifteen species to mitigate carbon dioxide

(CO$_2$) and urban pollutant concentrations, by analyzing the leaf-physiological traits (gas exchange) and morphological structures (stomata, trichomes, epicuticular waxes, and cuticular ornamentation) involved in pollutant removal. Their results suggested that the potential mitigation capacity, based on the investigated traits of the shrubs and herbaceous species, was species-specific. Speak et al. [62] quantified the effectiveness of four species, *Agrostis stolonifera*, *Festuca rubra*, *Plantago lanceolata*, and *Sedum albumat*, in capturing particulate matter smaller than 10 mm (PM10). The study found that the grasses, A. stolonifera and F. rubra, were more effective than *P. lanceolata* and *S. albumat* at PM10 capture.

In the Mediterranean climate, characterized by dry periods during the summer with high temperatures and solar radiation, it is necessary to provide an adequate quantity of water to the vegetation. It is necessary to know the response of the vegetation to conditions of water stress, thus identifying, in the substrate and in the drainage layer, the components capable of making water available to the vegetation without increasing the overall weight of the green roof. The ability of a plant species to withstand prolonged periods of water stress depends on their speed of transpiration, the water content in the substrate, and the resistance to the water transfer from the substrate to the vegetation. Savi et al. [63] analyzed the resistance to water stress of some plant species in the Mediterranean area, according to the indicator $\psi_{50}$ that is the water potential of substrate. Higher values of this parameter correspond to a greater capacity of a species to withstand intense and prolonged water stresses. Currently, the parameter $\psi_{50}$ is the most used, among physiological criteria, to select species suitable for green roofs. In the Mediterranean area, it is necessary to use species characterized by strongly negative critical values of $\psi_{50}$. These species use a greater quantity of water from the soil, enduring the increased transpiration, due to the high temperatures and increased solar radiation in the summer season.

### 5.7.2. Sedum Species

Numerous researchers have identified succulent plants as the species with the highest performance for extensive green roofs. Among these, Sedum species are the most common because of their capacity to reduce transpiration and store additional water in leaves, allowing them to withstand drought conditions. In addition, Sedum species exhibit crassulacean acid metabolism (CAM), which improves their water-use efficiency by permitting stomatal opening and CO$_2$ storage during the night-time, when evaporation rates are lower than during the day. However, Sedum species are unable to exploit additional water.

The most used Sedum species are the following:

- *Sedum sediforme* [64–66]
- *Sedum album* [67–69]
- *Sedum kamtschaticum* [67,70]
- *Sedum lineare* [71,72]

Nektarios et al. [66] evaluated the growth capacity of native *Sedum sediforme* in extensive systems. It was found that *Sedum sediforme* was able to survive under minimal or no irrigation even at the shallow depth of 7.5 cm and proved to be a native plant species that could successfully be utilised in extensive systems in the Mediterranean and other semi-arid climatic regions.

Several studies underlined the potential of Sedum species to survive extended periods without water. Nagase and Dunnett [73] investigated plant survival, following an imposed drought. The authors concluded that the drought tolerance of Sedum species was superior to that in forbs and grasses. Lu et al. [71] highlighted that Sedum species survived through a five-week continuous drought treatment. With a restricted water supply, a deeper substrate (no less than 10 cm) was recommended by authors to ensure better drought tolerance performance of the plants in extensive green roofs. Additionally, Sedum species were demonstrated to be effective for a shallow substrate. Eksi et al. [74] analyzed a pre-vegetated mat of a mixture of sedum on shallow substrates, with a depth of 5 cm.

### 5.7.3. Other Possible Plant Species

In humid tropical climates, *Portulaca grandiflora* has shown high performance, while, in warm and dry climates with long periods of drought, *Aptenia cordifolia* was proved to be suitable. Furthermore, upholstery plants offer greater performance, in terms of visual quality, compared to Sedum species, increasing the biodiversity in the greening. Synergy with succulent species is very significant, as upholstery plants can exploit an excess of humidity, without which they require additional irrigation, especially during the summer months. Blanusa et al. [75] used a range of contrasting plant types, such as a *Sedum mix*, *Stachys byzantina*, *Bergenia cordifolia*, and *Hedera hibernica* (Figure 4). The results showed that *Stachys* outperformed the other species, in terms of leaf surface cooling, substrate cooling below its canopy, and, even, cooling the air above its canopy. Gionannini et al. [76] compared the performance of plant communities with that of monocultures and compared the growth of natives to non-natives in a simulated green roof setting in a desert environment. Native plants selected were *Chrysactinia mexicana*, *Melampodium leucanthum*, *Euphorbia antisyphilitica*, and *Nassella tenuissima*, and non-natives were *Delosperma nubigenum*, *Stachys byzantina*, *Sedum kamtschiaticum* and *Festuca glauca*. The authors concluded that the lack of differences in plant performance regardless of assignment to monoculture or community would imply that communities and monocultures are equally suitable for an arid region.

Species mixtures have often been correlated to enhanced aesthetic value. Furthermore, plant variety could improve substrate cooling, prevent intrusive unwanted plants, and preserve water. Heim and Lundholm [77] tested three drought-tolerant, mat-forming species native to *Nova Scotia*, *Cladonia* spp. (lichen), *Polytrichum commune* (acrocarpous moss), and *Danthonia spicata* (bunchgrass). The results showed that the incorporation of functional diversity, especially varied growth forms, increases the diversity of green roofs, potentially improving the resilience and performance of these systems in the long term. Van Mechelen et al. [78] analyzed the initial plant composition of commercial extensive systems, in terms of functional diversity, and proposed two methods to compose species lists which maximize functional diversity. The authors believed that designing functionally diverse plant systems will support more sustainable urban planning and improve the quality of urban life.

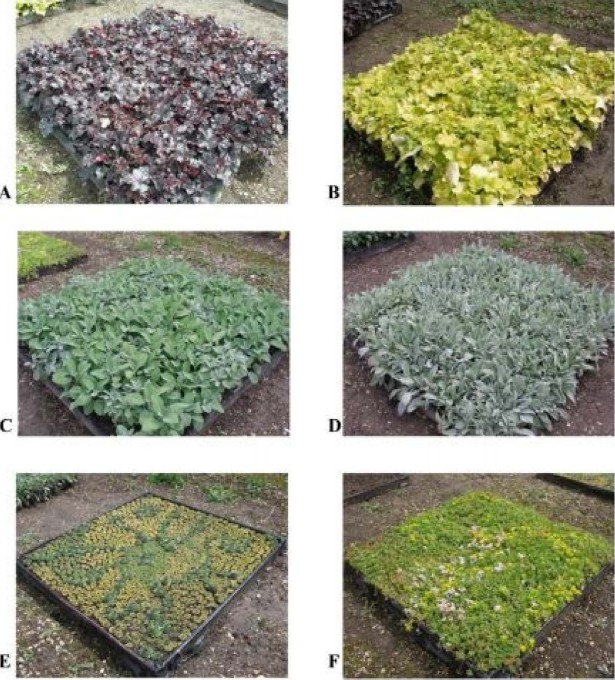

**Figure 4.** Plants used by Vaz Monteiro et al. [79]: A. *Heuchera 'Obsidian'*; B. *Heuchera 'Electra'*; C. *Salvia officinalis 'Berggarten'*; D. *Stachys byzantine*; E. *Sempervivum 'Reinhard'*; and F. A *Sedum* mix.

*5.8. Green Roof vs. Conventional/Traditional Roofs*

Vegetative roofs are gaining popularity, due to their many benefits compared to traditional roofs, since they absorb solar radiation and consequently waterproofing membrane is heated up by the sun during the day and cooled down at night. These daily temperature fluctuations could crack the roof membrane and reduce its durability.

El Bachawati et al. [80] focused on characterizing and analyzing the temperature profile of a traditional roof mockup and two extensive green roof mockups with different substrate depths and composition in the winter season. The traditional roof mockup consisted of the following layers: Roof assembly, thermal insulation layer, waterproof membrane, filter sheet, and an exterior layer made of pebbles (Figure 5). The green roof mockups were each installed using the following layers: Roof assembly, thermal insulation layer, waterproof membrane, root resistant barrier, drainage layer, filter sheet, growing medium, and vegetation. As for the substrate, it entailed oil, peat, alumina, pumice, and organic fertilizer. The vegetation layer was pre-cultivated elements.

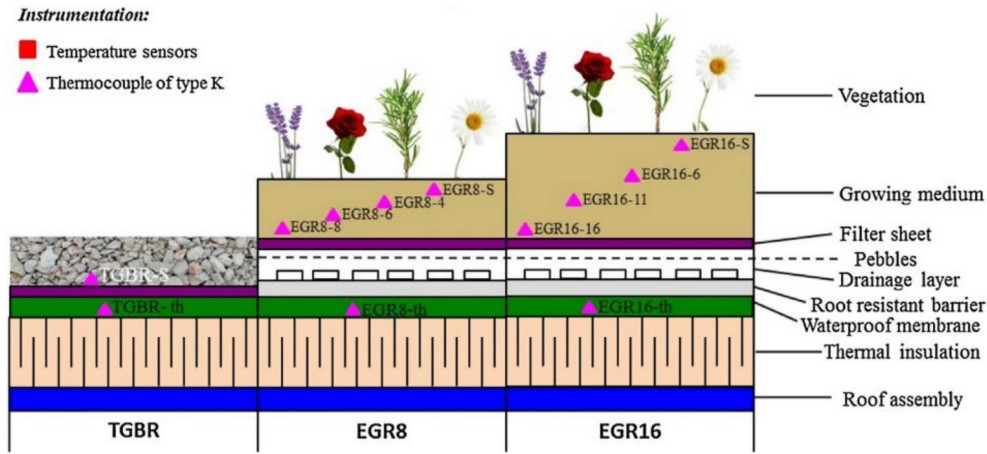

**Figure 5.** Comparison between traditional and green roof. (**a**) Traditional roof; (**b**) Extensive green roof (substrate thickness 8 cm); (**c**) Extensive green roof (substrate thickness 16 cm) [80].

Results from [80] showed that during warmer days, the substrate temperature was lower than that of the traditional roof surface. During colder winter days, daily average substrate temperature values were similar to that of the traditional roof surface, mostly due to partial plant coverage. The highest recorded temperature values were 26.18 °C for air, 33.98 °C for traditional roof surface, 24.24 °C for 8 cm substrate, and 23.36 °C for 16 cm substrate. This indicated that the highest temperature values of air and on traditional roof surface were greater than that of under substrates.

In addition, Dos Santos et al. [81] indicated that the use of green roofs resulted in lower temperature variations throughout the day, decreased internal temperatures, and decreased thermal amplitude in relation to a conventional roof (with tiles). The reduction was 0.8 °C, 1.7 °C and 1.6, respectively. Theodosiou et al. [82] showed that in Mediterranean countries, a green roof can contribute substantially to building's energy conservation mainly during the warm period of the year, while its influence during the cold period is negligible.

Since roof finishing materials of non-vegetated roofs are most often rigid, there is a large potential to attenuate sound waves diffracting over the outer skin of the building and to enhance quiet facades, e.g., in road traffic applications. Due to their relatively large surface mass density, low stiffness and pronounced damping properties, green roofs increase the acoustic insulation on top of the basic roof [83]. Especially their performance in the low-frequency range could be interesting. Van Renterghem, and Botteldooren [84] demonstrated that green roofs lead to consistent and significant sound reduction at locations where only diffracted sound waves arrive relative to common, non-vegetated roofs. A single

diffraction case with an acoustic improvement exceeding 10 dB was found for sound frequencies between 400 Hz and 1250 Hz, although the interaction path length was only 4.5 m.

In terms of cost, Sproul et al. [85] presented an economic perspective on roof color choice using a 50-year life-cycle cost analysis (LCCA) ND data collected from 22 flat roof projects in the U.S. The authors found that relative to black roofs, green roofs had negative net savings of $71/m$^2$ ($6.60/ft$^2$) because they cannot compensate for installation cost premium. However, owners concerned with local environmental benefits should choose green roofs.

## 6. Design Optimization for Mediterranean Climate

### 6.1. Influence of Climate Conditions

The majority of the regions with Mediterranean climates have relatively mild winters and very warm summers. Under the Köppen climate classification [86], "hot dry-summer" climates with average temperature in the warmest month above 22 °C (classified as Csa) and "cool dry-summer" climates with average temperature in the warmest month below 22 °C (classified as Csb) are often referred to as "Mediterranean". In many Mediterranean areas there is a strong diurnal character to daily temperatures in the warm summer months, due to strong heating during the day from sunlight and rapid cooling at night. As a result, areas with this climate receive almost all of their precipitation during their winter and spring seasons and may go anywhere from three to six months during the summer and early fall without having any significant precipitation [87].

Such conditions impose severe restrictions on plant growth and on plant survival and water is one of the most common limiting factors for the development of plants. Unfortunately, summer water shortages, a recurring problem in the Mediterranean region, are expected to increase according to climate change scenario forecasts. Yet irrigation is an unsatisfactory long-term option, both economically and ecologically. In Mediterranean regions high summer temperatures and prolonged seasonal drought make the installation of efficient and fully functional green roofs more difficult. Due to these climate restrictions, their design has been influenced; hence, new considerations about substrate characteristics and the plant species used are emerging in an adaptive approach to green roof construction in Mediterranean areas in contradistinction to the formalistic approach that is currently dominating the industry. It is therefore important to establish plant selection criteria prior to implementation. In the case of Mediterranean green roofs, the main criteria are drought tolerance, their indigenous nature, aesthetic characteristics (to ensure acceptance by the general public) and low maintenance requirements.

### 6.2. Possible Material Selection

Chenot et al. [88] assessed the role of substrate thickness and composition in maintaining the moisture necessary for good vegetation cover. The authors mixed fine elements (clays and silts) with coarse elements (pebbles of all sizes) with the aim of allowing typical pioneer Mediterranean vegetation communities to be maintained without human intervention (no watering, mechanical or chemical weeding). For the optimal colonization of the vegetation, the results showed that a substrate thickness of 15 cm composed mainly of clays and silt (75% clay-silt and 25% pebble-sand) would be recommended for the installation of green roofs with such substrates in a Mediterranean climate context. In two different papers, Monteiro et al. [89,90] tested the adequacy of different substrates for supporting aromatic plants. All experimental substrates proved to be adequate for vegetation growth, with the combination of 70% expanded clay, 15% organic matter and 15% crushed eggshell showing the best results regarding plant establishment and growth over time. These studies showed that selected aromatic plant species could be successfully used in green roofs in a Mediterranean climate. Ondoño et al. [91] studied the germination capacity and subsequent development of five Mediterranean species (*Silene vulgaris*, *Crithmum maritimum*, *Silene secundiflora*, *Lagurus ovatus*, *Asteriscus maritimus*, and *Lotus creticus*) on three different artificial substrates (green compost with crushed bricks, expanded clay and

clay–loam soil, respectively). The result showed that crushed bricks and expanded clay substrates were more appropriate for every plant species tested than the clay–loam soil mixture. The authors strongly recommended the use of lightweight and highly porous substrates as the basis for Mediterranean plants growth, and the combined use of perennial and annual species, such as *S. vulgaris* and *L. ovatus*, which offered a permanent cover throughout the year. The same research group, in a different paper [92], investigated the ability of four different substrates to maintain and promote the growth of two Mediterranean plant species. The two plant species tested in the above-mentioned substrates were *Lotus creticus* and *Asteriscus maritimus*. The plant species selected are good candidates to be introduced in green roof systems located in Mediterranean cities because, as we have demonstrated, both are perfectly adapted for growth under harsh weather conditions with little irrigation and low organic matter inputs. Raimondo et al. [93] provided insights into the importance of species-specific drought resistance strategies and hydraulic properties for selecting Mediterranean native species best suited for specific technical functions and ecological requirements of green roofs. Experiments were performed using two Mediterranean shrub species: *Arbutus unedo* and *Salvia officinalis*. Both vegetation types were found to be suitable species in the Mediterranean area.

As concern the substrate depth, Savi et al. [94] investigated the performance of two sub-Mediterranean shrubs (*Cotinus coggygria* and *Prunus mahaleb*) grown over green roofs with extremely shallow substrate depths and identified the impact of substrate thickness on shrubs water status, survival, and growth in a sub-Mediterranean climate. The results confirmed the possibility to install extensive green roofs vegetated with stress-tolerant shrubs in sub-Mediterranean areas using 10 cm deep substrate.

Van Mechelen et al. [95] provided an overview of plant traits that are crucial for the survival of plants in areas where dry periods are prominent, especially in the Mediterranean climate. The most important plant traits were incorporated in an easy to handle screening tool and it will be applied on a species list of a vegetation survey in Mediterranean southern France. The highest scoring species were Sedum album and Sedum acre, both already frequently used on green roofs. The authors highlighted that 35% of the species in the new potential species group recommended in the Mediterranean region are therophytes. Moreover, Caneva et al. [96] performed an extensive bibliographic search on plants proposed for extensive green roofs in Mediterranean countries, aimed at the creation of a wide database.

*6.3. Comparison with Tropical Climate*

A tropical climate is characterized by hot-humid summer with frequent showers, thunderstorms and occasional typhoons. A limiting factor for tropical green roof implementation is plant survival. Plant selection and testing for applications have taken place mainly in the Mediterranean climate, with a set of conditions that are radically different from those of the hot-humid tropics.

Jim [97] evaluated vegetation effect on green roof thermal energy performance with reference to climate adaptation in the compact tropical city of Hong Kong. From the findings, practical recommendations have been distilled to inform design, installation and maintenance of a simple, durable and low-maintenance green roof on building rooftops in humid-tropical cities. The author concluded that:

- Vegetation: Peanut has been found to perform significantly better than Sedum;
- Substrate: A 5 cm layer of soil composed of completely decomposed granite amended with 20% fully mature compost and slow-release fertilizer was suitable for Peanut growth;
- Rockwool layer: The rockwool layer had the benefit of lightweight and exceptionally high-water storage capacity which can enhance water supply to plants.

In another study, Jim [98] evaluated green roofs of three vegetation types with different growth forms and biomass structure in comparison with a control plot in a field-based study in a humid, tropical environment. The findings showed that grass cooled the air more than groundcover and shrub indicated the key role played by biomass quantity and structural complexity in molding the

passive cooling functions. Deng and Jim [99] established 94 voluntary vascular plant species from 26 families and 76 genera. They fall into three groups, namely dominant ruderal (herbaceous and sub-shrub) as a surrogate of early-stage local grassland ecosystem succession, arboreal (trees and shrubs), and hygrophilous herb. The results showed that local common ruderal plant species can be established and reproduced on a tropical extensive green roof.

## 7. Examples of Products Available in the International Market

Table 3 shows the technologies provided by green roof enterprises in Italy. It should be noted that the selected companies are case studies and there are more and more into the market. Most of them have headquarters in Europe and they only send the materials in Northern Italy. Almost all these enterprises provide extensive and intensive green roofs. Concerning vehicular and sloped vegetative roof, some technical precautions need to be installed. In particular, the sloped one requires special pieces in order to avoid the slipping of the substrate during rains and the excess of water runoff.

**Table 3.** Green roof technologies provided by enterprise case studies.

|  | Extensive | Intensive | Pedestrian Vehicular | PV | Sloped | Lightweight |
|---|---|---|---|---|---|---|
| Zinco | X | X | X | X |  |  |
| Bauder | X | X |  |  | X | X |
| Daku | X | X |  |  |  |  |
| Perlite | X | X |  |  |  |  |
| Harpo | X | X |  |  |  |  |
| Climagrun | X | X |  | X |  |  |
| Optigrun |  |  | X | X | X | X |

The majority of existing buildings date back prior to the entry into force of the laws regulating building energy consumption. Green roof technologies could be used to reduce energy consumption and to increase sustainability in buildings. however, it is necessary to determine overloading in relation to different configurations and to compare it with the residual load bearing capacity of the building structures. To avoid an expensive structural upgrade, some companies developed lightweight systems to keep the weight below the load limit prescribed by law.

A brief description of the commercial technologies in Table 3 is provided:

- Extensive: They are lightweight and have a shallow build-up height. Suitable plants include various Sedum species, herbs and some grasses. After the establishment of the vegetation, the maintenance is limited to one or two inspections a year.

- Intensive: They are usually multifunctional and accessible. They require more weight and a deeper system build-up. The maintenance is regular and depends on the landscape design and the chosen plant material. Anything is possible from lawns, perennials, shrubs, trees, including other landscape options, such as ponds, pergolas and patios.

- Pedestrian/Vehicular: During the installation of the different build-up layers the waterproofing has to be protected from damage. It is possible to install a protection mat or a drainage layer which functions as a protective layer as well. Driveways on rooftops require both a stable construction and adequate load-bearing capacity. Additionally, to the self-weight and imposed loads on driveways, horizontal forces and torsional movements may occur through acceleration, steering or breaking.

- Photovoltaic-green roof: The panels are covered with a prescribed amount of growing medium and the desired vegetation is then planted. The combined weight of the growing media and plants provides the ballast required by the solar energy system to deal with wind loads. Thanks to this ballast principle, roof membrane penetrations that would normally be necessary for anchoring standard solar energy systems are not required.

- Sloped: The plant selection has to be well adapted to the extreme conditions of steep pitched green roofs, where the solar radiation is the highest on the south facing roof side and the water runoff is much faster compared to a flat roof.
- Lightweight: It comprises mature sedum on 20 mm of extensive substrate and incorporating multifunctional water retention and filter layer. The system is suitable for both new build construction and retrofit refurbishment projects. In most instances an additional drainage layer is not required though on roofs up to 2° or in areas of high rainfall, its inclusion may be necessary.

The combination of green roofs with photovoltaic (PV) panels is a new tendency in the building sector because it provides synergistic benefits, such as the panel is cooled by the presence of the vegetation, and thus produces more electricity, while the solar panel enhances growing conditions for vegetation, and increases abiotic heterogeneity, resulting in higher plant diversity. In the Mediterranean area, where the annual average solar radiation and air temperature are high, several studies explored the possibility to combine the energy generation with extensive green roofs that are not walkable. Lamnatou and Chemisana [100] carried out a critical review about multiple factors which are related to PV-green roofing systems. The studies revealed that plant/PV interaction resulted in PV output increase depending on parameters, such as plant species, climatic conditions, evapotranspiration, albedo, etc. The same authors in another study [101] focused on the experimental evaluation of Photovoltaic (PV)—green roofs under Mediterranean climate summer conditions, demonstrating the benefits of this technology and filling the gap which existed in the literature in terms of the experimental evaluation of PV-green systems. The results obtained for a sunny, five-day time period revealed an average increase of the maximum power output of the PVs (ranging from 1.29% to 3.33% depending on the plant), verifying the positive synergy between the PVs and the plants. Schindler et al. [102] concluded that in a Mediterranean climate it would be appropriate to examine the use of irrigation in green roofs with PV panels, including effects on the plant community and on electricity production.

Table 4 compares materials and product used in green roof technology available into the international market.

**Table 4.** Materials and product available in the international market.

| Anti-Root Membrane | | | | | | |
|---|---|---|---|---|---|---|
| Parameters | N. 1 | N. 2 | | | | |
| Thickness (mm) | 1.1 | 0.36 | | | | |
| Surface mass (g/m$^2$) | 1130 | 310 | | | | |
| Breaking strength (N/5cm) | 80 | 20-47 | | | | |
| Breaking expansion (%) | >20 | >400 | | | | |
| **Drainage layer** | | | | | | |
| Parameters | N. 1 | N. 2 | N. 3 | N. 4 | N. 5 | N. 6 | N. 7 |
| Height (mm) | 45 | 19 | 25 | 40 | 60 | 75 | 25 |
| Surface mass (kg/m$^2$) | 2.0 | 19 | 1.7 | 2.0 | 2.2 | 1.0 | 5.0 |
| Resistance (kN/m$^2$) | 138 | 400 | 200 | 170-250 | 40-533 | 55 | 460 |
| Water storage (l/m$^2$) | 17 | - | 3.0 | 6.0 | 13 | 3.0 | - |
| Runoff 1% slope (l/(s × m)) | - | 0.34 | 0.59 | 1.5 | 1.1 | 1.54 | 1.0 |
| Runoff 2% slope (l/(s × m)) | - | 0.47 | 0.85 | 2.1 | 1.6 | 2.21 | 1.5 |
| Runoff 3% slope (l/(s × m)) | - | 0.57 | 1.05 | 2.6 | 2.0 | - | 1.9 |
| **Filter layer** | | | | | | |
| Parameters | N. 1 | N. 2 | N. 3 | N. 4 | N. 5 | N. 6 | N. 7 |
| Thickness (mm) | 7.0 | 17-20 | 5.0 | 6.0 | 0.6 | 1.7 | 1.0 |
| Surface mass (g/m$^2$) | 650 | 1500 | 470 | 850 | 100 | >300 | >150 |
| Water storage (l/m$^2$) | 7.0 | 12 | 5.0 | 4.0 | - | - | - |
| Breakthrough force (N) | - | 2300 | >2000 | >3500 | 1100 | 4300 | 2250 |
| **Substrate** | | | | | | |
| Parameters | N. 1 | N. 2 | N. 3 | N. 4 | N. 5 | | |
| Dry Volumetric weight (g/l) | 1000 | 1000 | 950 | 1000 | 1120 | | |

**Table 4.** *Cont.*

| Anti-Root Membrane | | | | | |
|---|---|---|---|---|---|
| Saturated Volumetric weight (g/l) | 1500 | 1500 | 1400 | 1400 | 1400 |
| Maximum water capacity (%) | 50 | 50 | 45 | 40 | 28 |
| Permeability (mm/min) | 0.3–30 | 0.3–30 | 0.3–30 | 0.6–70 | 60–400 |
| pH (CaCl$_2$) | 6.5–8.0 | 6.5–8.0 | 6.5–8.0 | 6.5–8.0 | 7.0–8.5 |
| Saline content (g/l) | <2 | <2 | <2.5 | <2.5 | <2.5 |
| Organic matter (g/l) | <90 | <90 | <90 | <65 | <40 |
| Compacting factor | 1.3 | 1.25 | 1.25 | 1.2 | 1.12 |

## 8. Irrigation Systems in Mediterranean Climate

The performance of a green roof is also measured in relation to the amount of irrigation it needs. In the southern European countries with Mediterranean climate, compared to those of northern Europe with continental climate where such coverage is experiencing a wide diffusion, it is necessary to install the irrigation system. Green roofs are generally seen as a desirable building element, providing numerous benefits where water availability does not restrict their implementation. However, most Mediterranean locations have long, dry summers, requiring irrigation to sustain vegetation throughout extended dry periods.

Numerous variables intervene in the availability of water for the vegetation survival, such as the average annual rainfall, the distribution of rains, the trend of daytime and nocturnal temperatures and the relative humidity of the air. It is necessary to make a strong irrigation in the initial period to allow the growth vegetation. After that, irrigation should be considerably decreased according to the type of green roof. The Rain Irrigation System is the oldest, simulating the rainfall through high-pressure water sprayers. This system is suitable for both large and small green roofs. A part of the water supplied evaporates, due to wind and heat, before it reaches the ground and the root apparatus. The Micro-irrigation System is more modern and based on providing small amounts of water with high frequency, near the plant roots. The driers located at the base of the stem allow the capillary distribution of water. This system reduces water losses, due to wind and evaporation.

Azeñas et al. [103] quantified the effect of irrigation water volume on the thermal capacity of a green roof system in a Mediterranean area. The modules with the 25% of potential evapotranspiration applied as limited irrigation reported lower heat flux values than the well-irrigated module (considered as 50% of potential evapotranspiration) in all seasons. Schweitzer and Erell [104] demonstrated that the water requirements of the plant species tested ranged from 2.6 to 9.0 L/m$^2$ per day. *Aptenia cordifolia* was the most efficient in its use of water, providing the highest cooling benefit per unit water required for irrigation. The authors concluded that it was hard to justify green roofs in such environments on the basis of their contribution to building energy conservation, although other benefits may nevertheless make them attractive.

## 9. Recycled Materials for Green Roof Layers

In recent years, the volume of recycled products has increased. Finding alternative uses for these materials in the construction sector represents one of the major challenges for the industries working in this sector. Therefore, it is always desirable to utilize local waste material for substrates, which can make the establishment of a green roof inexpensive.

Several studies have evaluated the performance of alternative recycled materials for substrates, in reducing the embodied energy required to construct a green roof and divert waste from landfills. Table 5 shows that crushed brick and construction waste are the most-used recycled material as an inorganic component in green roof substrates. Importantly, Chen et al. [105] tested a normal cultivated substrate with recycled glass. Recycled glass is a lightweight and porous material which improves pollutant absorption and water quality purification. This substrate performed well in the neutralization of acid rain, but did not significantly reduce the levels of other pollutants. The results showed that

materials like recycled glass generally have higher performance than natural ones and have the advantage of increasing sustainability by recycling waste materials.

**Table 5.** Recycled materials in the green roof substrate.

| Authors | Reference | Recycled Material Used | Main Findings |
|---|---|---|---|
| Bisceglie et al. | [106] | Waste of granular Autoclaved Aerated Concrete | The pH value of the water extract was of 7.23; the organic matter was less than 4.08; the apparent density was 459.2 kg/m$^2$; the demand for high water retention capacity was completely satisfied by the value of 222.62% of the mass of water absorbed relative to the mass of the dry sample. |
| Chen et al. | [105] | Recycled glass | It performed well in the neutralization of acid rain, but did not significantly reduce the levels of other pollutants. |
| Matlock and Rowe | [107] | Crushed porcelain and foamed glass | Substrate volumetric moisture content was generally greater in shale than in foamed glass or porcelain. |
| Eksi and Rowe | [108] | Crushed porcelain and foamed glass | Total plant coverage in both porcelain and foamed glass was equivalent to expanded shale on five of the six dates measured over two growing seasons. Substrate moisture and temperature were observed during the second season. The moisture content of both the porcelain and foamed glass was either equivalent to or greater than that of the expanded shale throughout the season. Subsurface temperatures were cooler in the porcelain and foamed glass than the expanded shale during the daytime for the majority of the second season. Variation in daily temperatures in the porcelain was significantly lower than the expanded shale when plant coverage was below 50%. |
| Molineux et al. | [109] | Inert construction waste material | Some of the alternative substrates are comparable to the widely used crushed red brick aggregate (predominantly found in commercial green roof growing substrate) for supporting plant establishment. For some materials, such as clay pellets, there was increased plant coverage and a higher number of plant species than in any other substrate. |
| Bates et al. | [110] | Crushed brick, crushed demolition aggregate, and solid municipal waste incinerator ash aggregate | Treatments with a high proportion of crushed brick in the growth substrate supported richer assemblages, with more species able to seed, and a smaller amount of Sedum acre. |
| Mickovski et al. | [111] | Inert construction waste material | The substrate mix containing recycled construction waste materials was adequate in supporting plant growth, was resistant to erosion and slippage and capable of providing good drainage. |
| Molineux et al. | [112] | Clay and sewage sludge, paper ash, and carbonated limestone | Particle density and loose bulk density results have shown all substrates to be classed as lightweight aggregates and leaching analysis has confirmed that all substrates perform within legal leachate limits for drinking water. |
| Farias et al. | [113] | Sieved waste | The new aggregate had low bulk density and increased water absorption and porosity. The thermographic camera results provided evidence that new aggregates had significant insulating properties and were suitable for use on green roofs. |

Only a few studies have assessed the performance of recycled materials used in the drainage layer of green roofs, mainly focusing on the performance and advantages of recycled rubber crumbs from tires in the drainage layer. A research group [114–116] at the University of Lleida (Spain) built three identical house-like cubicles, located in Puigverd de Lleida, where the only difference was the roof construction system. These researchers evaluated the energy consumption and thermal behavior of green roofs. The reference-case roof consisted of a conventional flat roof with thermal insulation while, in the other two cubicles, the insulation layer was replaced by a 9 cm deep layer of recycled rubber crumbs and pozzolana, respectively, as drainage layer materials of an extensive green roof. Both the cubicles showed less energy consumption (16.7% and 2.2%, respectively) than the reference case during warm periods, whereas they presented a higher energy consumption (6.1% and 11.1%, respectively), during heating periods. Furthermore, the insulating properties of rubber crumbs were tested and compared with the material performance of stone. A reduction of indoor temperatures between 2–5 °C during the summer and early autumn was found. Then, the hydrologic performance of the recycled

rubber granules was studied and compared with that offered by stone materials. No significant differences were found in the hydraulic conductivity when pozzolana was replaced with rubber crumbs, especially when small and half-particulate sizes were used. Finally, a life cycle assessment (LCA) was applied to compare the environmental impact. Extensive green roof with recycled rubber had a significantly lower environmental impact, compared to the non-insulated conventional roof (7% reduction) and compared to the other vegetative roof, with pozzolana drainage layer (6.7% reduction), and had a similar environmental impact than a conventionally insulated roof (2% increase).

## 10. Discussion

In this section a comparison on the cited literature is performed, as suggested by Besir and Cuce [117]. Most of the previous studies focused on both selecting suitable substrate and vegetation. As concern the substrate, previous studies were classified into two big groups, depending on whether the aim was at analyzing the performance or the composition. Relating to substrate mixture, Eksi et al. [57] suggested that the addition of 60 or 80% compost resulted in the greatest plant growth and fruit yields. On the other hand, some studies suggested that the existence of organic material in the substrate was a cause of pollutants in the green roof runoff. In addition, organic components, such as coco-peat, were demonstrated to improve, by 5.2 times, their initial weight in the highest water content. The German guidelines FLL [21] for green roofs indicate that the substrate should include only 4–8% and 6–12% organic matter by volume for extensive and intensive green roofs, respectively. In countries where vegetative roof technologies are not commercially available, customers may use locally accessible materials for this assembly, such as garden soil and composts. However, normal garden soil is not suitable, being made by skilled gardeners more experienced in traditional gardens than in green roofs. Specific weaknesses related to the use of garden soil are poor water and nutrient retention, increased weight, and local wildflower growth. Furthermore, the use of 100% local mixtures should also be prevented, as this reduces support of the vegetation, encourages the development of unwarranted weeds, raises roof weight during rainfall events, and endangers the endurance of the entire roof. Therefore, the growth medium should be appropriately engineered to accomplish the advantages of green roofs and the features suitable for an ideal growth substrate.

Sedums were considered among the best plant species for use on extensive green roof types [71]. Contrary to the conventional logic that plants with high transpiration rates are superior, the authors established that, during the summer months, the Sedum species outperformed the herbaceous ones [74]. However, as Sedum species are not available in numerous areas of the world, investigations were also focused on testing further plant species suitable for green roofs and numerous studies have recommended the use of various kind of vegetation for increasing the efficiency of green roofs [75,76].

Several studies have been conducted on the temperature regime of green roofs compared to traditional roofs to prove that vegetative roofs protect the roof membrane from extreme temperature fluctuations [80–82]. Their results confirmed that green roof protected the roof membrane from high temperature fluctuations.

Among the layers analyzed, the drainage layer plays a fundamental role, because it ensures an optimal balance between air and water content in green roofs and creates the conditions for vegetation growth by storing water, allowing excess water runoff, and ensuring aeration of the substrate and root system. While several authors proposed and evaluated the performance of substrate using recycled materials (see Table 5), only a few studies considered innovative solutions (mainly rubber crumb [114–116]) for the drainage layer.

Because a green roof is a load on the roof structure, it is important to keep the weight below the load limit prescribed by law, i.e., 200 kg/m$^2$ (about 1.96 kN/m$^2$) is the load that can be applied on the flat roofs of residential buildings according to the European standard [118]. In a previous study [119], the authors compared the weights of three different granular drainage materials. Perlite and expanded clay represent commercial drainage solutions, while rubber crumb is an innovative solution in the green roof market. among the drainage materials investigated, rubber crumbs had the highest density

values, as well as the highest values of thermal conductivity. Therefore, the main advantage of choosing rubber crumbs derives from their recycled origin.

## 11. Conclusions

This review paper analyzed the roles, requirements, performance, and materials of the layers of a green roof: The waterproof and anti-root membranes; the protection, filter, and drainage layers; the substrate; and the vegetation. In an engineered system, the role played by each component is well-defined and the optimal selection of each component depends on geographic location, in order to get the best outcomes from the green roof. A change in any of the mentioned components could alter its efficiency.

Future research on green technology should consider the peculiarities and availability of the materials in the area where the green roof is installed, replicating the same configuration in locations characterized by different climatic conditions can negate the positive effects of a green roof. These materials should come from the recycling of local agricultural waste to reduce costs and to improve performance and sustainability. Physical characteristics, such as thermal conductivity and inertia, maximum and minimum densities, specific gravity, hydraulic conductivity, and void index, of these recycled materials should be assessed. Finally, a life cycle analysis should be carried out to analyze the environmental impacts of these materials, also considering the recycling process.

**Funding:** This research was funded by "the Notice 5/2016 for financing the Ph.D. regional grant in Sicily" as part of the Operational Program of European Social Funding 2014–2020 (PO FSE 2014–2020).

**Conflicts of Interest:** The author declares no conflict of interest.

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
