# Peer review of "Green Roof Design: State of the Art on Technology and Materials"

_sustainability, doi:10.3390/su11113020_

Round 1
Reviewer 1 Report
The paper is much more better than in the first run. Two small things I like to update:
Minor correction for the 2nd run: 23.04. 2019
Line 611: Table 2. Recycled materials in green roof substrate. * the mentioned brand names are only examples, there are a number of further deliverer ….
Avoid only to make advertisement for a few …
Line 781: the first line is missing: .. FLL 2018, ed. Or the link …
https://shop.fll.de/de/english-publications/green-roof-guidelines-2018-download.html
Guidelines for the Planning, Construction and Maintenance of Green Roofs; 2018
23. Guidelines for the Planning, Construction and Maintenance of Green Roofs; 2018;
Author Response
Dear Editor and Reviewer, thanks for your suggestions and comments.
Response to Reviewer 1:
Point 1: Line 611: Table 2. Recycled materials in green roof substrate. * the mentioned brand names are only examples, there are a number of further deliverer …. Avoid only to make advertisement for a few …
Response 1: Thanks for the observation. The manuscript was revised as follows:
Now in the text lines: 708-709
It should be noted that the selected companies are case studies and there are more and more in green roof market.
Now in the text line: 720
Table 3. Green roof technologies provided by enterprise case studies.
Point 2: Line 781: the first line is missing: .. FLL 2018, ed. Or the link …
https://shop.fll.de/de/english-publications/green-roof-guidelines-2018-download.html
Guidelines for the Planning, Construction and Maintenance of Green Roofs; 2018
23. Guidelines for the Planning, Construction and Maintenance of Green Roofs; 2018;
Response 2: The author agrees with the reviewer’s comment. The reference was modified as suggested by the reviewer.
New reference
21. FLL, 2018. Guidelines for the Planning, Construction and Maintenance of Green Roofs.

Reviewer 2 Report
I would like to thank the author for the amendments.
However, it is my opinion that there is still a fundamental lack of depth and novelty in this paper for it to warrant publication.
As pointed out by the author (lines 49-73), there has been multiple studies in green roofs and green roof technology. This paper presents content that is rather similar to studies such as:
Vijayaraghavan, K. (2016). Green roofs: A critical review on the role of components, benefits, limitations and trends. Renewable and sustainable energy reviews, 57, 740-752.
Technical details of green roof systems are comprehensively laid out is literature such as:
Landschaftsentwicklung, F. (2008). Guidelines for the planning, construction and maintenance of green roofing: Green roofing guideline. Forschungsgesellschaft Landschaftsentwickung Landschaftsbau.
In terms of how the paper is written, the emphasis has been on finding relevant literature of components of green roofs. However, this is not sufficient. There should be discussion and comparison on the cited literature. For instance, the following paper conducted a review of green roof and green wall components and provided analysis of the studies found:
Besir, A. B., & Cuce, E. (2018). Green roofs and facades: A comprehensive review. Renewable and Sustainable Energy Reviews, 82, 915-939.
This is an important layer of information that is missing in this manuscript.
Climatic conditions of the Mediterranean, which I assume this paper is for, has not been comprehensively described and compared with other climates. More importantly, it is not clear how such climatic characteristics should factor into green roof design. For example, how is the Mediterranean climate different from the Tropical climate and how does it affect plant selection? The Author could have discussed this point using cite literature. This would have made the manuscript much more meaningful.
Author Response
Dear Editor and Reviewer, thanks for your suggestions and comments.
Response to Reviewer 2:
Point 1: As pointed out by the author (lines 49-73), there has been multiple studies in green roofs and green roof technology. This paper presents content that is rather similar to studies such as:
Vijayaraghavan, K. (2016). Green roofs: A critical review on the role of components, benefits, limitations and trends. Renewable and sustainable energy reviews, 57, 740-752.
Technical details of green roof systems are comprehensively laid out is literature such as:
Landschaftsentwicklung, F. (2008). Guidelines for the planning, construction and maintenance of green roofing: Green roofing guideline. Forschungsgesellschaft Landschaftsentwickung Landschaftsbau.
Response 1: Thanks to the revisions provided by editor and reviewers, the quality of this manuscript is higher than the first paper submitted. Several sections were added and many new references were considered. For example, the following new paragraphs were created following reviewer suggestions:
· 2. History and modern applications
· 5.8. Green Roof vs. Conventional/Traditional Roofs
· 6. Design Optimization for Mediterranean Climate
· 7. Examples of Products available into the International Market
· 8. Irrigation systems in Mediterranean climate
· 10. Discussion
Consequently, the aim of this paper was changed, providing new abstract and keywords, and the novelty compared to previous research was underlined.
Now in the text lines: 85-97
Differently from both the review carried out by Vijayaraghavan [19] and the international FLL guidelines [21], the novelty of this paper consists in comparing it to a conventional roof technology, in terms of both materials and thermal and economic performance, in assessing the Mediterranean climate conditions and their influence on green roof design, also comparing it with Tropical area and focusing on irrigation systems, in providing examples about the commercial materials and products available in the market and in analyzing innovative materials coming from recycled sources, as possible components. All these aspects related to green roof materials and technology are not fully described neither by previous articles nor by international guidelines. In addition, for each layer, the roles, requirements, performance, and materials are assessed. The information provided in this review paper will be useful for both researchers and designers to develop Mediterranean guidelines for selecting suitable components and materials during the design and installation phases.
First, the history and modern applications are discussed, in order to present a state of the art of this technology and their benefits and classification into extensive and intensive are described.
Point 2: In terms of how the paper is written, the emphasis has been on finding relevant literature of components of green roofs. However, this is not sufficient. There should be discussion and comparison on the cited literature. For instance, the following paper conducted a review of green roof and green wall components and provided analysis of the studies found:
Besir, A. B., & Cuce, E. (2018). Green roofs and facades: A comprehensive review. Renewable and Sustainable Energy Reviews, 82, 915-939.
This is an important layer of information that is missing in this manuscript.
Response 2: The author agrees with reviewer comment. A discussion section was created and added to the paper.
Now in the text lines: 833-877
10. Discussion
In this section a comparison on the cited literature is performed, as suggested by Besir and Cuce [125]. Most of the previous studies focused on both selecting suitable substrate and vegetation. As concern the substrate, previous studies were classified into two big groups, depending whether the aim was at analyzing the performance or the composition. Relating to substrate mixture, Eksi et al. [64] suggested that the addition of 60 or 80% compost resulted in the greatest plant growth and fruit yields. On the other hand, some studies suggested that the existence of organic material in the substrate was a cause of pollutants in the green roof runoff. In addition, organic components, such as coco-peat, were demonstrated to improve, by 5.2 times, their initial weight in the highest water content. The German guidelines FLL [21] for green roofs indicate that the substrate should include only 4–8% and 6–12% organic matter by volume for extensive and intensive green roofs, respectively. In countries where vegetative roof technologies are not commercially available, customers may use locally accessible materials for this assembly, such as garden soil and composts. However, normal garden soil is not suitable, being made by skilled gardeners more experienced in traditional gardens than in green roofs. Specific weaknesses related to the use of garden soil are poor water and nutrient retention, increased weight, and local wildflower growth. Furthermore, the use of 100% local mixtures should also be prevented, as this reduces support of the vegetation, encourages the development of unwarranted weeds, raises roof weight during rainfall events, and endangers the endurance of the entire roof. Therefore, the growth medium should be appropriately engineered to accomplish the advantages of green roofs and the features suitable for an ideal growth substrate.
Sedums were considered among the best plant species for use on extensive green roof types [79]. Contrary to the conventional logic that plants with high transpiration rates are superior, the authors established that, during the summer months, the Sedum species outperformed the herbaceous ones [82]. However, as Sedum species are not available in numerous areas of the world, investigations were also focused on testing further plant species suitable for green roofs and numerous studies have recommended the use of various kind of vegetation for increasing the efficiency of green roofs [83,84].
Several researches have been conducted on the temperature regime of green roofs compared to traditional roofs to prove that vegetative roofs protect the roof membrane from extreme temperature fluctuations [88–90]. Their results confirmed that green roof protected the roof membrane from high temperature fluctuations.
Among the layers analyzed, the drainage layer plays a fundamental role, because it ensures an optimal balance between air and water content in green roofs and creates the conditions for growth of the vegetation by storing water, allowing excess water runoff, and ensuring aeration of the substrate and root system. While several authors proposed and evaluated the performance of substrate using recycled materials (see Table 5), only few studies considered innovative solutions (mainly rubber crumb [122–124]) for the drainage layer.
Because a green roof is a load on the roof structure, it is important to keep the weight below the load limit prescribed by law, i.e. 200 kg/m2 (about 1.96 kN/m2) is the load that can be applied on the flat roofs of residential buildings according to the European standard [125]. In a previous study [126], the authors compared the weights of three different granular drainage materials. Perlite and expanded clay represent commercial drainage solutions, while rubber crumb is an innovative solution in the green roof market. among the drainage materials investigated, rubber crumbs had the highest density values, as well as the highest values of thermal conductivity. Therefore, the main advantage from choosing rubber crumbs derives from their recycled origin.
New reference
125. Besir, A. B.; Cuce, E. Green roofs and facades: A comprehensive review. Renewable and Sustainable Energy Reviews 2018, 82, 915–939, doi:10.1016/j.rser.2017.09.106.
Point 3: Climatic conditions of the Mediterranean, which I assume this paper is for, has not been comprehensively described and compared with other climates. More importantly, it is not clear how such climatic characteristics should factor into green roof design. For example, how is the Mediterranean climate different from the Tropical climate and how does it affect plant selection? The Author could have discussed this point using cite literature. This would have made the manuscript much more meaningful.
Response 3: Thanks for the comments. A new section, named “6. Design Optimization for Mediterranean Climate”, was edited and added to the manuscript. In this paragraph, the Mediterranean climate conditions were described, their influence on green roof design were assessed, and the results were compared to Tropical ones. Following these improvements suggested by the reviewer, this paper is now more meaningful.
Now in the text lines: 611-705
6. Design Optimization for Mediterranean Climate
6.1. Influence of Climate Conditions
The majority of the regions with Mediterranean climates have relatively mild winters and very warm summers. Under the Köppen climate classification [94], "hot dry-summer" climates with average temperature in the warmest month above 22 °C (classified as Csa) and "cool dry-summer" climates with average temperature in the warmest month below 22 °C (classified as Csb) are often referred to as "Mediterranean". In many Mediterranean areas there is a strong diurnal character to daily temperatures in the warm summer months due to strong heating during the day from sunlight and rapid cooling at night. As a result, areas with this climate receive almost all of their precipitation during their winter and spring seasons and may go anywhere from 3 to 6 months during the summer and early fall without having any significant precipitation [95].
Such conditions impose severe restrictions on plant growth and on plant survival and water is one of the most common limiting factors for the development of plants. Unfortunately, summer water shortages, a recurring problem in the Mediterranean region, are expected to increase according to climate change scenario forecasts. Yet irrigation is an unsatisfactory long-term option, both economically and ecologically. In Mediterranean regions high summer temperatures and prolonged seasonal drought make the installation of efficient and fully functional green roofs more difficult. Due to these climate restrictions, their design has been influenced; hence, new considerations about substrate characteristics and the plant species used are emerging in an adaptive approach to green roof construction in Mediterranean areas in contradistinction to the formalistic approach that is currently dominating the industry. It is therefore important to establish plant selection criteria prior to implementation. In the case of Mediterranean green roofs, the main criteria are drought tolerance, their indigenous nature, aesthetic characteristics (to ensure acceptance by the general public) and low maintenance requirements.
6.2. Possible Material Selection
Chenot et al. [96] assessed the role of substrate thickness and composition in maintaining the moisture necessary for good vegetation cover. The authors mixed fine elements (clays and silts) with coarse elements (pebbles of all sizes) with the aim of allowing typical pioneer Mediterranean vegetation communities to be maintained without human intervention (no watering, mechanical or chemical weeding). For an optimal colonization of the vegetation, the results showed that a substrate thickness of 15 cm composed mainly of clays and silt (75% clay-silt and 25% pebble-sand) would be recommended for the installation of green roofs with such substrates in a Mediterranean climate context. In two different papers, Monteiro et al. [97,98] tested the adequacy of different substrates for supporting aromatic plants. All experimental substrates proved to be adequate for vegetation growth, with the combination of 70% expanded clay, 15% organic matter and 15% crushed eggshell showing the best results regarding plant establishment and growth over time. These studies showed that selected aromatic plant species could be successfully used in green roofs in a Mediterranean climate. Ondoño et al. [99] studied the germination capacity and subsequent development of five Mediterranean species (Silene vulgaris, Crithmum maritimum, Silene secundiflora, Lagurus ovatus, Asteriscus maritimus, and Lotus creticus) on three different artificial substrates (green compost with crushed bricks, expanded clay and clay–loam soil, respectively). The result showed that crushed bricks and expanded clay substrates were more appropriate for every plant species tested than the clay–loam soil mixture. The authors strongly recommended the use of lightweight and highly porous substrates as the basis for Mediterranean plants growth, and the combined use of perennial and annual species, such as S. vulgaris and L. ovatus, which offered a permanent cover throughout the year. The same research group, in a different paper [100], investigated the ability of four different substrates to maintain and promote the growth of two Mediterranean plant species. The two plant species tested in the above-mentioned substrates were Lotus creticus and Asteriscus maritimus. The plant species selected are good candidates to be introduced in green roof systems located in Mediterranean cities because, as we have demonstrated, both are perfectly adapted for growth under harsh weather conditions with little irrigation and low organic matter inputs. Raimondo et al. [101] provided insights into the importance of species-specific drought resistance strategies and hydraulic properties for selecting Mediterranean native species best suited for specific technical functions and ecological requirements of green roofs. Experiments were performed using two Mediterranean shrub species: Arbutus unedo and Salvia officinalis. Both vegetation types were found to be suitable species in the Mediterranean area.
As concern the substrate depth, Savi et al. [102] investigated the performance of two sub-Mediterranean shrubs (Cotinus coggygria and Prunus mahaleb) grown over green roofs with extremely shallow substrate depths and identified the impact of substrate thickness on shrubs water status, survival, and growth in a sub-Mediterranean climate. The results confirmed the possibility to install extensive green roofs vegetated with stress-tolerant shrubs in sub-Mediterranean areas using 10 cm deep substrate.
Van Mechelen et al. [103] provided an overview of plant traits that are crucial for survival of plants in areas where dry periods are prominent, especially in the Mediterranean climate. The most important plant traits were incorporated in an easy to handle screening tool and it will be applied on a species list of a vegetation survey in Mediterranean southern France. The highest scoring species were Sedum album and Sedum acre, both already frequently used on green roofs. The authors highlighted that 35% of the species in the new potential species group recommended in the Mediterranean region are therophytes. Also, Caneva et al. [104] performed an extensive bibliographic search on plants proposed for extensive green roofs in Mediterranean countries, aimed at the creation of a wide database.
6.3. Comparison with Tropical Climate
Tropical climate is characterized by hot-humid summer with frequent showers, thunderstorms and occasional typhoons. A limiting factor for tropical green roof implementation is plant survival. Plant selection and testing for applications have taken place mainly in Mediterranean climate, with a set of conditions that are radically different from those of the hot-humid tropics.
Jim [105] evaluated vegetation effect on green roof thermal energy performance with reference to climate adaptation in the compact tropical city of Hong Kong. From the findings, practical recommendations have been distilled to inform design, installation and maintenance of a simple, durable and low-maintenance green roof on building rooftops in humid-tropical cities. The author concluded that:
· Vegetation: Peanut has been found to perform significantly better than Sedum.
· Substrate: A 5 cm layer of soil composed of completely decomposed granite amended with 20% fully mature compost and slow-release fertilizer was suitable for Peanut growth.
· Rockwool layer: The rockwool layer had the benefit of lightweight and an exceptionally high-water storage capacity which can enhance water supply to plants.
In another study, Jim [106] evaluated green roofs of three vegetation types with different growth forms and biomass structure in comparison with a control plot in a field-based study in a humid, tropical environment. The findings showed that grass cooled the air more than groundcover and shrub indicated the key role played by biomass quantity and structural complexity in molding the passive cooling functions. Deng and Jim [107] established 94 voluntary vascular plant species from 26 families and 76 genera. They fall into three groups, namely dominant ruderal (herbaceous and sub-shrub) as surrogate of early-stage local grassland ecosystem succession, arboreal (trees and shrubs), and hygrophilous herb. The results showed that local common ruderal plant species can be established and reproduced on tropical extensive green roof.
New references
94. Kottek, M.; Grieser, J.; Beck, C.; Rudolf, B.; Rubel, F. World Map of the Köppen-Geiger climate classification updated. Meteorologische Zeitschrift 2006, 15, 259–263, doi:doi:10.1127/0941-2948/2006/0130.
95. Peel, M. C.; Finlayson, B. L.; Mcmahon, T. A. Updated world map of the Koppen-Geiger climate classification. Hydrology and Earth System Sciences 2007, 4, 439–473.
96. Chenot, J.; Gaget, E.; Moinardeau, C.; Jaunatre, R.; Buisson, E.; Dutoit, T. Substrate composition and depth affect soil moisture behavior and plant-soil relationship on Mediterranean extensive green roofs. Water (Switzerland) 2017, 9, 1–16, doi:10.3390/w9110817.
97. Monteiro, C. M.; Calheiros, C. S. C.; Martins, J. P.; Costa, F. M.; Palha, P.; de Freitas, S.; Ramos, N. M. M.; Castro, P. M. L. Substrate influence on aromatic plant growth in extensive green roofs in a Mediterranean climate. Urban Ecosystems 2017, 20, 1347–1357, doi:10.1007/s11252-017-0687-9.
98. Monteiro, C. M.; Calheiros, C. S. C.; Palha, P.; Castro, P. M. L. Growing substrates for aromatic plant species in green roofs and water runoff quality: Pilot experiments in a Mediterranean climate. Water Science and Technology 2017, 76, 1081–1089, doi:10.2166/wst.2017.276.
99. Ondoño, S.; Martínez-Sánchez, J. J.; Moreno, J. L. Evaluating the growth of several Mediterranean endemic species in artificial substrates: Are these species suitable for their future use in green roofs? Ecological Engineering 2015, 81, 405–417, doi:10.1016/j.ecoleng.2015.04.079.
100. Ondoño, S.; Martínez-Sánchez, J. J.; Moreno, J. L. The inorganic component of green roof substrates impacts the growth of Mediterranean plant species as well as the C and N sequestration potential. Ecological Indicators 2015, 61, 739–752, doi:10.1016/j.ecolind.2015.10.025.
101. Raimondo, F.; Trifilò, P.; Lo Gullo, M. A.; Andri, S.; Savi, T.; Nardini, A. Plant performance on Mediterranean green roofs: Interaction of species-specific hydraulic strategies and substrate water relations. AoB PLANTS 2015, 7, doi:10.1093/aobpla/plv007.
102. Savi, T.; Boldrin, D.; Marin, M.; Love, V. L.; Andri, S.; Tretiach, M.; Nardini, A. Does shallow substrate improve water status of plants growing on green roofs? Testing the paradox in two sub-Mediterranean shrubs. Ecological Engineering 2015, 84, 292–300, doi:10.1016/j.ecoleng.2015.09.036.
103. Van Mechelen, C.; Dutoit, T.; Kattge, J.; Hermy, M. Plant trait analysis delivers an extensive list of potential green roof species for Mediterranean France. Ecological Engineering 2014, 67, 48–59, doi:10.1016/j.ecoleng.2014.03.043.
104. Caneva, G.; Kumbaric, A.; Savo, V.; Casalini, R. Ecological approach in selecting extensive green roof plants: A data-set of Mediterranean plants. Plant Biosystems 2015, 149, 374–383, doi:10.1080/11263504.2013.819819.
105. Jim, C. Y. Assessing climate-adaptation effect of extensive tropical green roofs in cities. Landscape and Urban Planning 2015, 138, 54–70, doi:10.1016/j.landurbplan.2015.02.014.
106. Jim, C. Y. Effect of vegetation biomass structure on thermal performance of tropical green roof. Landscape and Ecological Engineering 2012, 8, 173–187, doi:10.1007/s11355-011-0161-4.
107. Deng, H.; Jim, C. Y. Spontaneous plant colonization and bird visits of tropical extensive green roof. Urban Ecosystems 2017, 20, 337–352, doi:10.1007/s11252-016-0596-3.

Reviewer 3 Report
This review paper discusses the green roof technology as an important element to improve the energy performance of a building. The paper is well articulated and organized analysing the various technological layers and materials. Starting from some historical introduction, the contemporary elements are discussed providing materials and other studies’ results.
Anyway, some improvements are necessary to consider the paper for publication.
First of all, for a deeper and better comparative discussion, the various technological elements should be compared to an ordinary (typological) roof technology. That will allow to better understand the improvements produced by a green roof. Some drawings would help in understanding the technological differences. Also a comparison in costs would be valuable to assess the construction feasibility. Finally, more technical prescription should be provided, such as in lines 175-177; 303-307; 695-696; etc.
Secondly, the paper is mainly focused in the Mediterranean area but, as a review, and also following the provided examples (France, Germany, etc. uses), a discussion of materials or products available in the international market would give a wider view of the topic. It would be important to actualise the provided data until the present days (i.e. lines130-137).
Thirdly, the energy improvement discussion (as a central theme of this paper/technology) should be deepened providing data and analysing the differences with an ordinary roof technology. Some words and the sound insulation should also be provided.
Then, tables 2 and 3 have the same title. A more suitable caption must be provided. In both the tables, a new column with a brief description of the technology (table 2) and the main findings per reference (table 3) would help in summarising the results.
In lines 527-532, the indicator Ψ50 should be described to make all the readers understand its meaning. Furthermore, it must be written always in the same way.
The word “review” must be removed from the keywords.
Finally, the English should be improved. “Green roof” is obsessively repeated making the reading boring and difficult. A native speaker revision is recommended.
Author Response
The response to the reviewer’s comments includes Figure and Tables, therefore it is not possible to enter it in this box, please see the Word file uploaded.

Round 2
Reviewer 2 Report
Thank you for improving on the paper. I have no other comments
This manuscript is a resubmission of an earlier submission. The following is a list of the peer review reports and author responses from that submission.
Round 1
Reviewer 1 Report
Green roof design: state of the art on technology and materials
This paper looks into different components of green roofs and provides information on their performance and practical implementation considerations, based on available literature.
In general, I find that this paper is a general literature review paper. Most information can be found in green wall design guidebooks. There is no new knowledge arising from this study.
On the topic of green roofs, this paper only covers a cursory review of all related components, without sufficient depth and discussion.
For instance, there is no in-depth analysis or critique on the actual extensive green roof systems commonly used today. This includes tray systems, trays with reservoirs, roll-up mat systems, hybrid solar panel and green roof systems, etc. This is a significant difference that impacts water storage and plant selection. Also, the author almost entirely misses out the topic of irrigation, which is another significant aspect of green roofs. Anyone who has conducted an experiment with green roofs will know that these are the more significant parameters to look into. The amount of rigor in describing the various components leave much to be desired.
Therefore, it is my opinion that the paper is not suitable for publication.
Reviewer 2 Report
I am not completely convinced, what this paper shall present:
On one hand there are much more publication on the market then these cited ones.
On the other side: I understand the paper as an overview about factors, which had to be mentioned, if regional guidelines should be developed …. Such as FLL 2018.
We know, that there some Green roof guidelines and norms exist in several countries in Europe, but also similar in America or Asia.
The paper did not cover all key publication, which are available on the market now.
This would be an argument to reject the paper.
I only could accept the paper, if the aim is more focussed, e.g. to present the key aspect for forming local/ national green roof guideline.
In such case, I recommend also to add the aspect of Retention rates, provided by different systems.
About the title: State of the art, I miss a little bit the history, what is current technology, and please present a little bit clearer what you recommend for future – better green roofs.
The selected literature is focussed on Italy/ south Europe. Perhaps it would be helpful to set a focus on green roofs in the dry or Mediterranean climate of the world. …
The paper has weaknesses, needs major improvements.
Some details:
Line 70: What does “practical green roofs mean?”
Line 353: FLL must be integrated as a citation: The newest Green roof guideline is dated 2018.
Line 493: Sedum is stuff enough for an own chapter; which species from which origin. Several cultivars performs best for EGRs.
Why such high standard for Green roofs related to the local plant species. Similar discussion we can do for garden plant selection …
It is always a question about the purpose: To get a well flowering roof garden – this is a difference to the nature habitat planning.
Line 500ff: it is difficult to conclude the natives for some regions:
Bergenia cordifolia from Asia
Stachys byzantinica: from Mediterranean …
Sedum different species from various origin around the globe.
This discussion is very tricky:
Green roofs are man-made habitats, Why not using suitable garden plants?
In very special projects, a local focus on natives can be a planning aim – that demands a very serious local search on the fittest for such purpose.
This is finally too difficult for such a short chapter.
Line 556: recycled materials as drainage layer: in most cases too heavy.
Line 601: Similar installation in various climate regions for compare: The big green roof companies realized green roof projects with similar material in various climate regions.
A survey of projects of these companies could be classified e.g. by regions, age, building types. …
To the editors:
If this paper shall be understand as “guide” to develop local green roof standards – ok it can be accepted.
If this paper should present an review “State of the art” I miss key publication and see massive weaknesses, e.g. in the chapter of plant species, this is real tricky. We can accept green roof plantings as a type of landscape architecture.
If it shall fulfil requirements about local and native plants – this needs more in depth citation or a local focus.
Reviewer 3 Report
I think some references are needed for parts: since line 134 to 144 ; 235-239;
Line 22-223 are you sure it is 3cm? are you sure is for water storage? please control and make example
3.3 and 3.4 have the same title
Substrate title is 3.4 as the previous paragraph
Performance title has number 3.1.1 but it is a mistake and so for the following paragraphs
Examples in Recycled maerials: I know that those of Lleida case are different papers but the experiment cubicles I think are always the same, if it is so explaine, it is very strange to read three times of these cubicles